# Intranasal Delivery: Effects on the Neuroimmune Axes and Treatment of Neuroinflammation

**DOI:** 10.3390/pharmaceutics12111120

**Published:** 2020-11-20

**Authors:** Elizabeth M. Rhea, Aric F. Logsdon, William A. Banks, Michelle E. Erickson

**Affiliations:** 1Geriatrics Research Education and Clinical Center, Veterans Affairs Puget Sound Health Care System, Seattle, WA 98108, USA; alogsdon@uw.edu (A.F.L.); wabanks1@uw.edu (W.A.B.); mericks9@uw.edu (M.E.E.); 2Division of Gerontology and Geriatric Medicine, Department of Medicine, University of Washington School of Medicine, Seattle, WA 98195, USA

**Keywords:** intranasal, neuroimmunology, neurodegenerative disease, cytokine, growth factor, vitamin

## Abstract

This review highlights the pre-clinical and clinical work performed to use intranasal delivery of various compounds from growth factors to stem cells to reduce neuroimmune interactions. We introduce the concept of intranasal (IN) delivery and the variations of this delivery method based on the model used (i.e., rodents, non-human primates, and humans). We summarize the literature available on IN delivery of growth factors, vitamins and metabolites, cytokines, immunosuppressants, exosomes, and lastly stem cells. We focus on the improvement of neuroimmune interactions, such as the activation of resident central nervous system (CNS) immune cells, expression or release of cytokines, and detrimental effects of signaling processes. We highlight common diseases that are linked to dysregulations in neuroimmune interactions, such as Alzheimer’s disease, Parkinson’s disease, stroke, multiple sclerosis, and traumatic brain injury.

## 1. Introduction

A current working definition of neuroimmunology can be that it is the field interested in the ways that the immune and nervous systems are connected. The field has a diversity of roots that have conglomerated over the decades to now cover an exceptionally wide range of interests and applications; indeed, it is increasingly difficult to think of a brain condition that does not have a neuroimmune component. Hans Selye [1] is often mentioned as one of the earliest pioneers of the field, whose work highlighted the relation between stress and disease, incorporating the role of the hypothalamic–pituitary adrenal axis (HPA). Early work by the pioneers revealed phenomena that still astound such as the work of Ader and Cohen, a psychologist and an immunologist, who worked together, showing that responses induced by toxins could be entrained or learned in Pavlovian fashion [2]. In the late 1970s and early 1980s, the role of new agents such as the cytokines showed that immune cells do more than synthesize antibodies and kill bacteria. The finding that immune agents are secreted not just by immune cells, but by a diversity of other cells has connected the three great integrating systems of the body: the immune system, the nervous system, and the endocrine system. Earlier names for the field are psychoneuroimmunology, psychoendocrinology, and psychoimmunology. Even the newest name, immunepsychiatry, fails to capture the diversity of phenomena that the field now encompasses, from placebo effects, to connections of psychosis with immune responses, to mechanisms for neurodegenerative diseases, to mediation of effects on the brain from the microbiome, to sickness behavior [3,4,5,6]. The term neuroimmunology is increasingly used in recognition that the underlying phenomena are at least as important to understanding normal physiology as disease states.

The blood–brain barrier (BBB) is an important dimension of neuroimmunology. Classically, the role of the BBB has been to separate the central nervous system (CNS) and the immune system. A widely held view by majority science until recently was that this separation of the immune system and CNS broke down only in disease states. While the BBB does prevent unregulated interactions, it is now understood that it plays a central role in regulated interactions and participates in the formation of a number of neuroimmune axes and interactions [7]. Indeed, the BBB is better envisioned as a blood–brain interface, in which the barrier function is only one of several of its important characteristics [8]. It is also a target of neuroimmune and neuroinflammatory events and is involved in pathological and adaptive responses in Alzheimer’s disease (AD), Parkinson’s disease (PD), multiple sclerosis (MS), stroke, and traumatic brain injuries (TBIs).

The complexities of the BBB make it the biggest hurdle to the delivery of therapeutics to the CNS and the treatment of brain diseases. As such, routes bypassing the BBB are of great importance. Intranasal (IN) delivery is one such alternate route to the CNS. Here, we consider the characteristics of the intranasal route of delivery for modifying neuroinflammation.

## 2. Intranasal Delivery Overview

IN administration can provide a route for delivering substances to the CNS. It offers advantages over intravenous (IV) or other routes of administration in a number of circumstances: (1) it may have a higher degree of delivery to the CNS compared with other routes of administration; (2) it can target the CNS, avoiding systemic side effects as typically only a small amount enters the blood stream; (3) it avoids degradation by serum proteases; and (4) it is relatively non-invasive. Most therapies have first shown pre-clinical beneficial effects following direct injection into the CNS using intracerebroventricular delivery (ICV) or after peripheral administration, but only benefit from one of the three advantages noted above [9].

Here, we will focus on the benefits after adaptation to the IN method. The literature reviewed for this manuscript is comprised of studies from rodents, non-human primates, and humans. Therefore, we will briefly introduce and summarize some of the literature regarding different IN delivery technicalities for each species. IN delivery as reviewed here refers to the technique of administration that favors direct uptake into the brain. This is distinct from two other objectives for IN administration, that of treating conditions within the nasal passages, such as sinusitis, or for delivering a substance into the blood stream to deliver an effect on a peripheral tissue. The techniques of administration and sites of application differ for these three different objectives.

For IN delivery targeting the CNS, the directed site of application is the cribriform plate. As the rodent nasal cavity differs anatomically from the human nasal cavity, the delivery techniques vary. In rodents, there are two common ways to deliver substrates by the IN route. The first, most common method of delivery is to drop the substrate (≤20 µL for mice and up to 50 µL for rats) using a pipette tip on the edge of the nares that is then inhaled [10,11,12]. This method is often repeated on the opposite nares and can be administered over time to reach the target concentration without administering too much volume in any single injection. Modifications to this method have found delivery can be enhanced by closing the mouth and opposite naris to allow for natural inhalation high into the cavity [13]. In addition, a recent Journal of Visualized Experiments (JoVE) video showed pressing the mouse head down and forward following this delivery method reduces the amount delivered to lungs [14]. This same method can be used in conscious, awake rodents by restraining the rodent via a scruff restraint, but also requires acclimation to the handling [15]. Another modified form of delivery in anesthetized rodents that is readily used is to insert a pipette tip or tubing shallowly into the nasal cavity [16,17,18]. The substrate is then delivered to the cribriform plate to allow for greater direct delivery to the CNS, limiting the amount absorbed by the turbinates or swallowed and absorbed by the gastrointestinal tract. The distance to insert the pipette tip has been determined by injecting Evan’s blue or similar dye, pushing beyond the turbinates and not puncturing the cribriform plate. With this method, typically only about 4–8% of the IN dose of a water-soluble substance enters the bloodstream. This modified method of delivery also allows the dose available to the CNS to be calculated, as well as the percent of the administered dose entering the brain and blood and the tissue concentrations to be achieved.

In non-human primates, larger volumes can be administered (0.1–0.5 mL/nares), typically closer to the upper level of the nasal passage near the cribriform plate through the use of flexible tubing (25–30 mm insertion), similar to the latter method described above for rodents [19,20]. Another mechanism of delivery is by spraying the substrate of interest in a mist through a controlled, pulsed pressurized atomizer, held in front of the animal’s nostril, similar to the method used in human clinical trials described below [21,22].

In humans, IN delivery is often delivered with nebulizers, sometimes called atomizers, that create pressured air to convert the liquid into a fine spray of aerosols. This method can improve deposition to the upper narrow part of the nose compared with other spray pumps. Delivery is alternated between nostrils with a designated interval between each administration to allow sufficient time for absorption [23,24]. Successful use of delivery devices is critical to the therapeutic outcome [24].

Brain entry after IN administration likely occurs by getting into the cerebrospinal fluid (CSF) at the cribriform plate as well as retrograde transmission at the olfactory and trigeminal nerves [25]. Other routes seem to be needed to explain characteristics such as the nearly instant distribution and equilibrium of a substance throughout the brain. Early studies found delivery primarily to the olfactory bulb, but subsequent studies have found substances that are more evenly distributed or even with the highest percent uptake by the hippocampus or hypothalamus. An observation without much follow-up is that distribution among brain regions can be modified by co-administration of cyclodextrins [26,27], suggesting that targeting specific brain regions is possible.

The subsequent distribution after IN delivery greatly depends on the biological and pharmacokinetic properties of a substance. For a water-soluble substance with no brain-to-blood efflux and rapid degradation in blood, exemplified by insulin, the substance is largely confined to the CNS. For a small lipid soluble substance, exemplified by progesterone, distribution can be throughout the body with tissue concentration determined by other factors, such as localized sequestration and degradation.

To complement our previous review investigating the IN delivery of proteins and peptides in the treatment of neurodegenerative diseases [9], we have chosen to focus this review on neuroinflammation that can be targeted with IN delivery. For these substances, we have grouped the treatments into six major categories: growth factors, vitamins and metabolites, cytokines, immunosuppressants, exosomes, and lastly stem cell therapy (Figure 1). We have summarized the biochemical and behavioral findings from these pre-clinical studies and clinical case reports in Table A1.

## 3. Growth Factors

### 3.1. IGF-1/Insulin

Insulin-like growth factor-1 (IGF-1) has neuroprotective properties, including protection against global neuronal loss and ameliorating hippocampal damage following cerebral ischemia. IN delivery of IGF-1 was first characterized in 1999 in rats and appears in the CNS within 20 min of administration, with the highest levels present in the olfactory bulb [28]. This pharmacokinetic study helped provide the feasibility to pursue this treatment for various disorders affecting the brain including stroke [29], depression and anxiety behaviors [30], and neuroinflammation [31].

Following transient middle cerebral artery occlusion (MCAO), three repeated doses of IN IGF-1 (37.5 µg and 150 µg) were administered within a 48 h period [13]. Only the greater dose was able to reduce infarct volume and improve motor function. ICV lipopolysaccharide (LPS) in newborn rodents is a model of perinatal infection and inflammation that can lead to brain injury in newborns, as well as long-term consequences in children and adults. IN IGF-1 can protect against ICV LPS-induced injury in the developing rat brain, including a decrease in the amount of microglia and polymorphonuclear neutrophils (PMN) [32]. Ultimately, this therapy prevented the loss of catecholaminergic neurons to improve motor function [31].

It should be noted the beneficial effects of IGF-1 within the CNS depend on how much IGF-1 reaches the brain. That is, high levels of IGF-1 in the CNS can be detrimental [33,34], while other studies have shown a beneficial effect of a higher dose of IGF-1 compared with the low dose [13]. These studies also highlight that the neuroprotection offered by IGF-1 may also depend on animal models used as well as the pathologic condition investigated [33]. Therefore, the route of administration (i.e., ICV, IN, IV) should be taken into consideration for each disorder.

The effects of insulin on inflammation were investigated both in the periphery as well as the CNS. Delivery of insulin following various routes of administration and the impact on the CNS has been recently reviewed [35]. Insulin was first shown to decrease inflammation in the periphery by decreasing IL-1β and TNF-α [36]. However, more recent studies have shown an anti-inflammatory role for insulin in the CNS [37,38]. In particular, IN insulin can alter inflammatory pathways in the hippocampus of an AD mouse model including the T cell receptor signaling pathway, cytokine–cytokine receptor interaction, and cell adhesion molecule pathway as measured by RNA sequencing [39]. IN insulin treatment also decreased activation of astrocytes and microglia in the hippocampus of an ICV-streptozotocin rat model of AD [40,41] and can improve memory in a mouse model of human AD [42].

Biliverdin reductase A (BVR-A) is a pleiotropic protein acting as a kinase, transcription factor, and antioxidant. It is regulated by the insulin receptor and impaired in human postmortem AD hippocampus [43]. BVR-A has also been shown to attenuate inflammation [44,45]. BVR-A decreased toll-like receptor (TLR)4, interleukin (IL)-1β, IL-6, and tumor necrosis factor (TNF)-α in the brain of germinal matrix hemorrhage (GMH)-induced inflammation [45]. In addition, endothelial nitric oxide synthase (eNOS) suppression inhibited the beneficial effect of BVR-A because translocation to the nucleus required eNOS-derived nitric oxide. IN insulin was not only able to improve memory in a mouse model of AD, but also prevented the impairment of BVR-A activation that occurs in AD, which improved insulin signaling in the hippocampus and decreased oxidative stress [46]. Based on the evidence described above, IN insulin regulation of BVR-A activity is dependent on nitric oxide and involved in ameliorating neuroinflammation associated with AD.

IN insulin reduces hippocampal lesion volume and reduced hippocampal microglia activation, but not astrocyte activation following a mouse model of moderate TBI [47]. Brain injury induced by controlled cortical impact (CCI) did not alter IN insulin delivery [47], similar to what has been shown for AD [48].

### 3.2. FGF

Fibroblast growth factors (FGFs) are a class of cell signaling proteins crucial to normal development. FGF2 is also known as basic FGF and can bind heparin. IN FGF2 increased neurogenesis in the subventricular zone of the mouse brain [49] and improved spatial memory as well as decreased hippocampal degeneration in an AD rat model [50]. FGF2 ultimately ameliorates neuroinflammation through microglial CD200 signaling, which has been shown to be modulated by damage-associated molecular patterns (DAMPs), pathogen-associated molecular patterns (PAMPs), and amyloid β [51].

FGF21 is a metabolic hormone primarily produced by the liver with pleiotropic roles and provides a beneficial effect in a high-fat diet obesity model [52] and inflammatory-induced depressive-like behavior [53]. In a mouse model of human PD, FGF21 administered by the IN route every other day for 2 weeks improved the behavioral deficit, dopaminergic neuron function, and mitochondrial function, and decreased pro-inflammatory cytokine expression, concomitant with decreased astrocyte and microglial activation [54].

Tolerability and safety of IN fibroblast growth loop dimer (FGLL) has been performed on young, healthy male subjects as a way to bypass the BBB and deliver FGF to brain [55]. FGLL is an artificial ligand of the FGFR1 and the specific binding site for neural cell adhesion molecule (NCAM), a neural surface protein important for development and maintenance of the nervous system. FGL can stimulate neurite outgrowth and enhance synapse formation and function, in addition to enhancing memory and protecting against ischemia [56,57,58]. FGL is detected in the developing rat CSF 10 min following IN delivery and remains elevated up to 5 h after delivery [58]. Using a model known to induce human AD-like dementia, the impact of IN FGL was investigated [59]. IN FGL improved short-term memory and decreased the deposition of endogenous amyloid β, tau hyperphosphorylation, microglial and astrocyte activation, and neuronal cell death in the hippocampus. The timing of IN FGL (either before AD-like signs had appeared or later when AD-like signs were clearly developed) did not affect the ultimate benefit. In addition, while FGL treatment did not affect the total amount of GSK3β, a kinase that is dysregulated in AD, there was a significant increase in the phosphorylation of this protein, thus rendering it inactive. Sodium hyaluronate conjugated to FGL was able to enhance delivery of IN FGL and suppress IL-1β, IL-6, and TNFα expression in a model of hypoxic-ischemic encephalopathy in postnatal rats [60].

### 3.3. Brain-Derived Neurotrophic Factor

Brain-derived neurotrophic factor (BDNF) is a growth factor member of the neurotrophin family. Neurotrophins are crucial in promoting neurogenesis during development, but can also play a neuroprotective role in the adult brain, even in the aged brain [61]. BDNF is thought to play a major role in neuroplasticity, neurorepair, and cognition. Transgenic animal models of AD have shown an inverse correlation between BDNF levels and cognitive performance [62]. Decreased BDNF levels in brain have been associated with clinical depression [63,64], PD [65], and AD [66].

Restoring BDNF levels in patients with CNS disease remains an intriguing therapeutic option. BDNF is highly expressed in brain, but can also be found in peripheral tissues. In 1998, Pan et al. showed that BDNF can completely cross the BBB through a saturable transport system [67]. Although transport of BDNF across the BBB favors CNS therapy, peripheral degradation and a large volume of distribution result in poor sustained delivery to brain, greatly limiting BDNF as a therapeutic. The use of IN delivery of BDNF has been shown to be a promising alternative. It can reduce depression-like phenotype in rats [63]. In 2014, Chen and colleagues advanced the CNS delivery of BDNF, while reducing peripheral confounds, by combining IN delivery of BDNF with focused ultrasound (FUS). This technique was then applied to improve the outcome in a rat model for human PD [68].

Exercise has been shown to substantially increase BDNF levels in both rodents [69] and humans [70]. Studies are ongoing to determine whether exercise can improve disease outcomes by enhancing BDNF levels in the CNS. It has been shown that intense exercise can augment endogenous BDNF levels and attenuate demyelination in a mouse model for human MS [71]. Moreover, exercise-induced BDNF was associated with improved cognition in multiple rodent AD models [72] and in humans with mild cognitive impairment [73]. As such, these data not only advocate for exercise to treat CNS disease, but also for the option to supplement BDNF, perhaps through an IN route, to further improve disease outcomes.

Inflammatory insults to the brain, such as those induced by TBI and stroke, have been associated with altered BDNF levels. After TBI, BDNF levels decrease in both the brain [74] and in the blood, which may predict injury severity [75,76]. Moreover, circulating BDNF levels are decreased in patients with stroke, and especially those patients with post-stroke depression [77] and anxiety [78]. In 2011, Jiang et al. successfully administered BDNF into the brains of stroked rats through the IN route. They found no change in post-stroke infarct volume, however, IN delivery of BDNF rescued neurons and altered the neuroinflammatory profile [79]. These results suggest that BDNF may be neuroprotective through immunomodulatory mechanisms and that the IN route of delivery could be a viable therapeutic option to treat brain injury.

### 3.4. Nerve Growth Factor (NGF)

Nerve growth factor (NGF) is a neurotrophin that aids in neuronal growth, differentiation and survival, and recovery following injury of brain cells [80]. Pre-clinical studies have shown radioactively labeled NGF enters the CNS and is distributed throughout the brain [81]. NGF was also successfully delivered to the CNS via IN administration, and shown to reduce edema and cell death in rats exposed to TBI [82]. A case report recently showed IN NGF in an infant can improve severe neurological impairment due to late-onset group B Streptococcus (GBS) meningitis [83]. GBS meningitis can cause disruption of the BBB [84] and release neuroinflammatory signals [85]. This was the first study to show IN NGF could improve both cerebral functions and clinical conditions, meriting further investigation to fully understand the neuroprotective mechanisms of NGF. In another case report, IN NGF has been shown to improve neurological impairment following TBI in a 4-year old boy [86]. NGF was detected in the CSF and there were increased protein levels of doublecortin, a marker of neurogenesis, following treatment.

### 3.5. Pituitary Adenylate-Cyclase-Activating Polypeptide

Pituitary adenylate-cyclase-activating polypeptide 38 (PACAP38) is a potent neurotrophic factor and has neuroprotective effects [87]. β-cyclodextran was able to increase IN PACAP38 delivery to regions including the frontal cortex, hippocampus, and hypothalamus [26]. In addition, a single administration of IN PACAP38 improved memory in aged SAMP8 mice, a mouse model of AD. Extended treatment with IN PACAP38 for 3 months in a mouse model of AD decreased the amount of soluble amyloid β; decreased gene expression of receptor for advanced glycation end products (RAGE), a protein involved in chronic inflammatory diseases, diabetes, and AD; and improved memory in the novel object recognition test [88]. IN administration of a PACAP analog for 3 months (5×/week) can promote polyglutamine-expanded androgen receptor (polyQ-AR) degradation, by decreasing phosphorylation, leading to destabilization, and improve neurotoxicity associated with spinobulbar muscular atrophy (SBMA) [89]. In a mouse model of Huntington’s disease (HD), IN PACAP was able to promote expression of hippocampal BDNF and decrease the formation of mutant huntingtin aggregates [90]. This is despite the decrease in the primary receptor for PACAP, PAC1, found in the hippocampus of HD patients as well as a mouse model of HD. In addition, 7 days of IN administration led to improvements in both short- and long-term memory. Synaptic density as measured by PSD95 was improved with IN treatment.

The improvement of many neuroinflammatory markers in a host of diseases mentioned here in both pre-clinical and clinical case studies warrants further investigation and careful consideration for the use of IN growth factors in ameliorating detrimental immune-related effects due to disease.

## 4. Vitamins and Metabolites

### 4.1. Vitamin A/Retinoic Acid

All-trans-retinoic acid (RA) is a potent bioactive derivative of vitamin A and regulates many cellular functions throughout the body during development and in adulthood. Vitamin A is an essential micronutrient that is usually obtained in sufficient quantities from a well-balanced diet. The metabolism of vitamin A to retinoids contributes to its cellular uptake, transport, and storage. RA and other bioactive retinoids signal in an autocrine or paracrine manner to nuclear receptors that include retinoic acid receptors (RARs) and retinoid X receptors (RXRs). There are three subtypes of RARs (RARα, β, and γ) and RXRs (RXRα, β, and γ), and each subtype can be processed into two or more isoforms via alternative splicing [91]. RARs and RXRs dimerize as homo or heterodimers to modulate transcription of genes with retinoic acid response elements (RAREs), and expression of RAR and RXR subtype combinations varies depending on cell type [92], leading to a high complexity of retinoid signaling mechanisms.

In general, RA acts to control proliferating cells by inhibiting their division and directing their differentiation [93]. It is a critical regulator during embryonic development, including neurodevelopment [94,95], and regulates neural plasticity, circadian rhythms, neurotransmission, and metabolism in adults. Another function of retinoids is the regulation of immunity. This is clearly exemplified by the fact that increased infections are a common consequence of vitamin A deficiency [92]. During development, retinoids are essential for the proper development and differentiation of the thymus, as well as secondary lymphoid organs such as Peyer’s patches in the intestines [92]. In adults, RA has immunomodulatory effects on dendritic cells, T-cells, B-cells, and macrophages that are highly dependent on cell-type and inflammatory context [92,96]. RA also regulates neuroimmune responses. In experimental autoimmune encephalomyelitis (a rodent model of MS), pre-treatment with RA dramatically reduced the clinical severity of disease. This effect was attributed to impaired dendritic cell maturation, and associated reductions in inflammatory monocytes and inhibited polarization of Th1/Th17 cells, ultimately leading to reduced inflammatory demyelination in the spinal cord [97]. Following middle cerebral artery occlusion, a mouse model of stroke, prophylactic treatment with i.p. RA reduced infarct volume, neurodegeneration, pro-inflammatory cytokine expression, and neutrophil infiltration [98]. However, RA does not always confer protective effects, highlighted by a recent study that investigated RA in a mouse model of systemic lupus erythematosus (SLE). Although RA had previously been shown to protect against kidney disease in SLE, oral RA treatment in the SLE model surprisingly exacerbated neuroinflammation and neurodegeneration [99].

Retinoid therapies are commonly used to treat severe forms of acne and other skin conditions and may be administered as topical creams or systemically. However, systemic treatment of retinoids can lead to many complications including skin and mucosal issues, altered circulating lipids such as increased low-density lipoprotein (LDL) and triglycerides and decreased high-density lipoprotein (HDL), hepatotoxicity, musculoskeletal problems, and teratogenic effects [100]. Therefore, IN delivery of retinoids may be a strategy to more specifically target retinoid signaling in the brain while mitigating systemic effects. Two recent studies have investigated the potential therapeutic applications of the IN retinoid 9-*cis* retinoic acid (9-*cis*-RA) in rodent models of stroke and amyloid β-driven AD pathology. 9-*cis*-RA differs from RA in that it potently activates RXRs, whereas RA has relatively weak action on RXRs [101,102]. 9-*cis*-RA was also shown to have greater neuroprotective potency versus RA [103]. It was shown that IN application of 20 µg 9-*cis*-RA in a volume of 20 µL DMSO in rats can increase brain levels to 10 nM concentrations after 1 h, which is sufficiently potent to activate RXR signaling [104,105]. In a rat model of stroke, IN 9-*cis*-RA administered on days 3–5 following MCAO improved dark cycle motor deficits versus vehicle at 7 and 14 days post-injury, and better motor functions were associated with increased neurogenesis in the subventricular zone [104]. Interestingly, RA treatment had an opposite effect on lowering vertical activity in sham rats, indicating that IN RA effects are context-specific. In mice harboring APP/PS1 mutations, which cause the overexpression and deposition of human amyloid β in the brain, it was shown that a 4 week regimen of IN 9-*cis*-RA given at 20 µg every other day reduced plaque deposition in the cortex and hippocampus as well as the expression of GFAP by astrocytes in the brain. 9-*cis*-RA also increased the cortical and hippocampal levels of the post-synaptic marker PSD95, suggesting that RA prevents synaptic loss in this model. Finally, it was shown that 9-*cis*-RA reduced the protein levels of the pro-inflammatory cytokines IL-1β, IL-6, and TNF-α [106]. Taken together, these data suggest that intranasally-applied retinoids may have therapeutic activities against neuroinflammation and neurodegeneration. However, much more work needs to be done exploring systemic activities of IN retinoids, as well as the beneficial versus detrimental effects that may be a consequence of dosing route, cellular targets, concentration, relative activity at RAR versus RXR receptors, and inflammatory context.

### 4.2. Vitamin D/Calcitriol

Vitamin D is comprised of hydrophobic compounds transported in blood bound to carrier proteins. Vitamin D3 is the precursor to the most active form of vitamin D, calcitriol, and is metabolized by the liver and kidney [107]. While vitamin D is most commonly known for its role in calcium and bone metabolism, it is also known for its anti-inflammatory actions [108]. Activation of the vitamin D receptor (VDR), which acts as a nuclear transcription factor, leads to activation of neuroprotective signaling pathways.

IN administration of an active metabolite of vitamin D3 and potent agonist of the VDR, calcitriol, reduces BBB disruption in a rat model of subarachnoid hemorrhage by increasing endogenous expression of astrocytic osteopontin (OPN), a neuroprotective glycoprotein [109]. IN vitamin D3 prevented the detrimental cleavage of CD44, an OPN receptor and adhesion molecule with diverse biological processes, including inflammation [110]. This was suggested to also restore P-glycoprotein (P-gp)—an important efflux transporter at the BBB that is often altered under CNS disease conditions—levels back to sham level.

Because of the anti-inflammatory nature of vitamin D, this vitamin has been proposed to aid in combating COVID-19 by (1) preventing SARS-CoV-2 infection, (2) preventing the cytokine storm associated with COVID-19, and (3) preventing the loss of neural sensation by stimulating expression of neurotrophins like NGF [111]. IN delivery could particularly aid in point 3 by acting directly at the site of action in olfactory neurons.

### 4.3. Vitamin E/α-Tocopherol

Vitamin E is comprised of numerous lipophilic compounds and many neurological diseases are associated with levels. One metabolite of vitamin E, α-tocopherol, is able to prevent neurotoxicity (i.e., astrocyte activation) [112]. Vitamin E has been studied for the prevention of oxidative stress in a rodent model of PD [113]. Vitamin E intake is higher in age-matched healthy controls compared with people with PD [113]. Therefore, patients with PD could already be at a disadvantage to benefit from this vitamin.

Vitamin E is often used as a compound in nanoemulsions because of its lipophilic nature. It also has the added benefit of antioxidant activity. As lipophilic nanoemulsions aid in CNS delivery following IN administration, many studies have shown beneficial effects using various nanoemulsions containing vitamin E. One example used IN naringenin (NRG), a potent flavonoid, loaded vitamin E nanoemulsions, which improved behavior in a pre-clinical model of PD [114]. While oxidative stress was reduced in brain, the full extent of the effect on neuroinflammation was not studied.

### 4.4. Other

DL-3-n-butylphthalide (NBP) is a family of compounds derived from celery seeds and has been shown to be a multifaceted beneficial drug for the treatment of stroke, hypertension, and neurodegenerative diseases [115]. IN NBP increased neuro- and angiogenesis accompanied by an increase in BDNF, vascular endothelial growth factor (VEGF), and eNOS following TBI [116].

While we have presented the findings from individual vitamins and their metabolites, there are plenty of other protective vitamins that have not been investigated following IN treatment. With differences in lipid solubility, it is important to keep in mind how the substrate may be distributed throughout the brain, as mentioned in the IN delivery section. In addition, the combinatorial effect of vitamins or metabolites on neuroinflammation following IN delivery has not been studied.

## 5. Cytokines/Anti-Inflammatory Agents

### 5.1. TLR Signaling

CNS insults, such as stroke and TBI, can activate TLRs, which then stimulate pro-inflammatory signaling responses such as nuclear factor kappa beta (NF-κβ)-dependent cytokine production. IN administration of a TLR/NF-κβ pathway inhibitor reduced the infarct volume and improved neurological function in a rat model of stroke [117]. Transforming growth factor beta (TGF-β) is an anti-inflammatory cytokine that can attenuate NF-κβ signaling [118,119]. IN delivery of TGF-β was shown to increase neurogenesis and improve the outcome in a mouse model of stroke [120].

Granulocyte-colony stimulating factor (G-CSF) is another cytokine that is upregulated by TLR signaling that can function to activate the innate immune response and promote cell proliferation [121]. G-CSF is highly upregulated in a variety of disease states and inflammatory conditions and is thought to be neuroprotective [122]. Its levels peak in blood about 12 h after TBI, making it a potential diagnostic biomarker [123].

IN delivery of G-CSF enhanced angiogenesis and improved the outcome in rats subject to ischemic stroke [124]. These studies suggest that IN delivery of cytokines that modulate the immune system and promote cell proliferation may be a viable treatment option for stroke and other CNS diseases associated with neuroinflammation.

### 5.2. Interferons

MS is a disease characterized by robust neuroinflammation and CNS immune cell infiltration [125,126]. Interferons are cytokines released as a host response to viral invasion to modulate the immune system. The interferon response has been harnessed and employed as an FDA-approved treatment for MS [126]. Interferon β (IFN-β) can block the activation and CNS recruitment of T lymphocytes to decrease neuroinflammation associated with MS [127]. Preclinical data show that IN delivery of interferons is distributed throughout the brain at a much higher level than following IV administration [128], and may prove to be a more efficacious clinical treatment for MS. Only about half of MS patients respond to interferon treatment [129], and perhaps this is because of the heterogeneity of BBB function. While IN delivery of interferons could also be considered for the clinical treatment of other CNS diseases associated with the adaptive immune response, care should be taken as interferons also have harmful or off-target side effects.

### 5.3. Anti-Inflammatory Cytokines: IL-13, IL-10

Endogenous cytokines can influence both the innate and adaptive immune systems. Interleukin 13 (IL-13) is an anti-inflammatory cytokine that is released by T helper (Th)2 cells [130]. IL-13 can signal through the IL-13 receptor a1 and has been shown to activate both endothelial cells [131] and neurons [132]. Studies have shown that IL-13 plays an important role in immune cell recruitment under inflammatory conditions, and that Th2 production of IL-13 is elevated in the CSF of relapsing MS patients [133,134].

IL-13 has been shown to associate, and even enhance the expression of, growth factors such as BDNF [135]. As described above, IN delivery of neurotrophic factors, like BDNF, can treat CNS diseases; although an alternative approach would be to administer cytokines that enhance endogenous activity of neurotrophic factors within the CNS. Recently, it has been shown that IN delivery of IL-13 can reduce neuroinflammation and improve the outcome in a rodent model of TBI [136]. IN delivery of anti-inflammatory cytokines may be an effective strategy to enhance neurotrophic factors and improve CNS disease.

T lymphocytes are a major regulator of IL-10 signaling and this process has been shown to be involved in LPS-induced depressive-like behavior. Major depressive disorder is often associated with alterations in the immune system. IN delivery of a neutralizing anti-IL-10 antibody led to upregulation of Ido1 in the prefrontal cortex, an amino acid enzyme that is positively correlated with depression scores in humans. In addition, the neutralizing antibody led to prolonged depression-like behavior, showing IL-10 can improve behavior [137]. These data suggest IN delivery of IL-10 could be beneficial in upregulating this protective pathway.

### 5.4. Pro-Inflammatory Cytokines: TNF-α

TNF-α is a potent pro-inflammatory cytokine that signals through one of two receptors, TNFR1 and TNFR2. TNF-α also exists in two bioactive forms, one that is membrane-bound and thus signals through cell–cell contacts, and the other that is soluble and generated through proteolytic cleavage of the membrane-bound form [138]. In general, membrane-bound TNF is thought to mediate many of the beneficial activities of TNF-α through TNFR2 signaling, whereas the soluble TNF-α form mediates pro-inflammatory activities through TNFR1 [139]. However, both receptors offer distinct protective functions in the brain [138]. TNF-α also contributes to physiological CNS functions that include strengthening of AMPA-receptor dependent excitatory neurotransmission [140] and regulation of sleep [141]. However, TNF-α is associated with pathological neuroinflammatory changes in neurodegenerative diseases, which have been reviewed by others [140]. TNF-α inhibition has been a successful therapeutic strategy for autoimmune diseases such as rheumatoid arthritis, plaque psoriasis, Crohn’s disease, and ulcerative colitis [140,142]. FDA-approved TNF-α inhibitors are biologics, and include monoclonal antibodies (infliximab, adalimumab, golimumab, certolizumab) and a recombinant fusion protein of the extracellular domain of the TNF-α receptor 2 fused to the human Fc domain of IgG (entanercept) [142].

TNF-α receptor inhibiting IgG antibodies can cross the BBB depending on the fusion proteins present [143]. In addition, soluble TNF-α has a saturable transport system at the BBB [144], and increased serum levels of TNF-α have been associated with increased rates of cognitive decline in AD patients [145]. A recent randomized, double-blind, placebo-controlled trial evaluated secondary outcomes of cognition, behavior, and global function following a weekly dose of subcutaneous entanercept. The trends in cognitive tests suggested that patients receiving entanercept performed better than placebo by the end of treatment, although the results were not significant and require cautious interpretation because of the small sample size [146]. Entanercept treatment was also well-tolerated without serious adverse events or new safety issues in this population. Perispinal and intrathecal administration of TNF-α inhibitors in a small, open-label study and a case report, respectively, showed pronounced cognitive improvements in dementia patients [147]. A recent randomized, double-blind control trial also showed that perispinal entanercept provides significant improvements in post-stroke pain [148]. Despite these apparent benefits, TNF-α inhibitors can increase the risk of demyelinating and non-demyelinating inflammatory CNS events in patients being treated for autoimmune diseases [142]. The proposed mechanisms for this paradoxical relation include reduced TNF-α mediated apoptosis of CNS autoreactive T-cells, TNF-α mediated regulatory T-cell survival, and perhaps increases in TNF-α in the brain [142,149], which would not be blocked by anti-TNF-α therapies because they do not reach effective concentrations in the brain. It remains to be determined whether neuroinflammatory side-effects extend to populations without autoimmune diseases.

Another therapeutic approach for lowering TNF-α in the brain is delivery of RNAi therapeutics. Yadav et al. have recently shown that IN pre-treatment with anti-TNF siRNA either in physiologic solution or incorporated into a cationic nanoemulsion was taken up into brain regions, and inhibited neuroinflammation induced by LPS injection in the substantia nigra [150]. Incorporation into cationic nanoemulsions improved brain uptake, with about 1.3% of the injected siRNA dose detected in the midbrain after 6 h of dosing versus about 0.3% of the unincorporated siRNA. Only the siRNA formulated in nanoemulsion significantly attenuated LPS-induced TNF-α expression in the brain, indicating that the treatment regimen reached therapeutic concentrations. Future studies are needed to determine the utility of this therapeutic approach for inhibiting the adverse effects of TNF-α. Whereas siRNA knockdown can be advantageous regarding genetic specificity, knockdown strategies for TNF-α would result in both the soluble and membrane-bound forms of TNF-α being reduced, and so both the pathological and physiological functions of TNF-α would be lost.

## 6. Immunosuppressants

### 6.1. Rapamycin

Fabio Di Domenico and Marzia Perluigi have repeatedly investigated the impact of IN rapamycin in a mouse model of Down syndrome [151,152]. Down syndrome individuals have an increased risk for AD, including cognitive decline and development of amyloid β plaques, neurofibrillary tangles, and neuroinflammation [153]. Mammalian target of rapamycin (mTOR) signaling is aberrant and hyperactive in Down syndrome [154] and AD. Rapamycin is an inhibitor of mTOR and is used as an immunosuppressant in organ transplant. Four hours following a single IN injection of 1 µg rapamycin, brain concentrations were 5.0  ±  1.0 ng/g, with the plasma concentration being 6.7  ±  1.3 ng/mL [152]. In contrast, a 50-fold greater i.p. injection of rapamycin resulted in just over twice as much appearance in the brain, with plasma levels being well over 100 times greater. This suggests systemic side effects can be lessened by delivering rapamycin intranasally versus via the i.p. route, while still maintaining similar brain levels. Following 12 weeks of IN treatment (3 × per week) of vehicle or 1 µg rapamycin, memory was improved in a mouse model of Down syndrome. IN rapamycin reduced hippocampal mTOR complex 1 (mTORC1) hyper-phosphorylation, without affecting peripheral mTOR signaling. IN rapamycin was shown to decrease hippocampal APP processing, amyloid-β deposition, tau hyper-phosphorylation, and tau kinases. Using the same experimental set-up as the study above, improvements in oxidative stress were found following IN rapamycin [151].

Another protein shown to inhibitor mTOR activity, Sestrin2, a stress-inducible protein, when delivered intranasally, improves stroke outcome (decreased infarct volume, improved cognition) following hypoxia-ischemia [155]. Therefore, targeting the mTOR pathway in the CNS through IN delivery could be a way to reduce the detrimental effects not only due to Down syndrome and stroke, but also other neurological diseases that have increased mTOR activity.

### 6.2. Cyclosporine-A

Cyclosporine-A (CSA) is a cyclic decapeptide that is a potent immunosuppressive agent with great neuroprotective properties and can act as a P-glycoprotein (P-gp) inhibitor. It can prevent neuronal damage due to excitotoxicity and regulate neurotransmitter release in addition to inducing production of neurotrophic factors [156]. IN delivery of CSA distributed throughout the brain via the olfactory neuronal pathway and trigeminal nerve pathway. In addition, IN delivery decreased CNS inflammation in response to LPS administration.

P-gp is an important efflux transporter located at the luminal surface of the BBB. It limits the ability of drug molecules to reach the brain. When P-gp substrates such as verapamil and talinolol were delivered intranasally, whole brain concentrations were greater than when these substrates were delivered IV, even without administration of the P-gp inhibitor CSA [157]. Therefore, IN delivery of known P-gp substrates is a therapeutic way to bypass the BBB and avoid BBB efflux.

## 7. Exosomes

Exosomes are a type of extracellular vesicle (EV) that are about 50–200 nm in diameter. They arise from endosomes and have important biological functions, such as regulating immune responses [158]. Exosomes can also be used as drug-delivery vehicles, and exosomes can cross the BBB [159,160,161]. Therefore, delivery of exosomes and their contents to the brain reflects a novel therapeutic strategy that could mitigate harmful neuroinflammatory processes [159]. Exosomes have been used to successfully deliver substances to the brain via the IN route in rodents. For example, tumor cell line-derived exosomes loaded with curcumin and delivered intranasally reduced induction of CNS IL-1β through a mechanism that involved apoptosis of reactive microglia, and the same exosome treatment also protected against the development of clinical symptoms in an experimental autoimmune encephalomyelitis (EAE) model [162]. Another group tested whether IN administration of catalase-loaded exosomes derived from macrophages could protect against neurodegeneration in a PD model that induces neurodegeneration in the substantia nigra via injection of 6-hydroxydopamine. Brain uptake of exosomes loaded with fluorescent dye in this model was evaluated following IN and IV injection, and it was found that more dye was taken up when exosomes were administered via the IN route [163]. This study showed that IN exosomes loaded with catalase reduced reactive microgliosis and protected against dopaminergic neuron loss in the substantia nigra. These two studies highlight that IN administration of exosomes could be a logical therapeutic delivery mechanism. Another possible advantage of the IN versus IV route could be lower uptake by peripheral organs. It was shown that IV injection of exosomes led to about 0.1% of the injected dose being in the brain 10 min following injection, whereas accumulation in the heart, liver, spleen, lung, and kidney was 1–2 orders of magnitude higher [159]. Systemic distribution of exosomes following IN administration has not yet been well-characterized. Brain-to-blood efflux mechanisms can contribute to exosome distribution, and some proteins that efflux from brain such as tau and α-synuclein [164] are detectable in CNS-derived exosomes [161,165]. Together, these studies highlight exosomes as attractive candidates for IN drug delivery to the brain, but more could be learned from understanding how features such as their cellular source/composition affect their biodistribution and function.

## 8. Stem Cells

Stem cell therapy has been proposed as a viable therapeutic option for the treatment of CNS disease [166,167,168]. Initially, researchers attempted to deliver stem cells directly to the CNS to initiate their neuroprotective effects. Mesenchymal stem cells (MSCs) are commonly delivered to the CNS via the IN route and have shown beneficial effects for CNS diseases such as TBI [169], stroke [170], PD [171], and brain cancer [172]. Although this therapeutic method shows promise in rodents, human safety concerns have cautioned clinical use [173,174,175]. In order to provide a safer alternative, researchers have attempted to harness the beneficial protective effects of stem cell delivery, while avoiding the detrimental proliferative response inherent to their machinery [176].

MSCs and other stem cells can be stimulated to secrete anti-inflammatory and neurotrophic factors [177], which can then be isolated and administered directly to the CNS via IN delivery [178]. This process has had preclinical success and researchers are now attempting to determine which secreted factors contribute to the beneficial effects of stem cell delivery [179]. MSC-derived EVs have garnered recent interest for their immunomodulatory role in treating CNS disease [176,177,180]. EVs encompass two types of vesicles: exosomes as described above and larger microparticles that are often upwards of 1000 nm in size and bud off from the cytoplasmic membrane. IN delivery of MSC-EVs attenuated microgliosis and dendritic spine loss in the 3×FAD mouse model for human AD [181]. MSC-EVs were shown to enter the rodent brain via the IN route and incorporate into microglia and neurons [182]. Preconditioning of MSCs with hypoxia [183] or inflammatory cytokines [181] can augment EV production and enhance their therapeutic efficacy. MSCs can also be incubated with factors that prime their EV production for IN delivery. For example, MSCs incubated with the Rho-kinase inhibitor, fasudil, were delivered intranasally and attenuated dopaminergic neuron loss in a mouse model for human PD [184]. Moreover, MSCs can also be genetically engineered to express various growth factors with neuroprotective properties [185].

MSCs can also be differentiated into CNS cell-types as a viable therapeutic strategy. For instance, conditioned media from MSC-derived oligodendrocytes was found to reduce inflammation and increase myelination in a rodent model for human MS [186]. Novel methods of stem cell incubation prior to IN delivery should be considered to both enhance CNS delivery as well as improve patient outcomes. Special emphasis should be considered to which CNS cell-types are dysfunctional in the disease of interest to determine differentiation and incubation parameters for the study.

## 9. Conclusions

In this review, we have highlighted the IN delivery of many different classes of compounds, focusing on neuroinflammation in CNS diseases including stroke, TBI, AD, PD, Huntington’s disease, and MS. Our review showed that IN is a viable route for the delivery to the brain on neuroimmune modulating substances. Administration by the IN route prevents systemic side effects and alteration of systemic inflammation. Sustained neuroinflammation can disrupt both the structure and function of the BBB [7,187,188]. Therefore, addressing and treating neuroinflammation is not only critical to improving the disease outcome, but also necessary to maintain a functioning BBB.

## Figures and Tables

**Figure 1 pharmaceutics-12-01120-f001:**
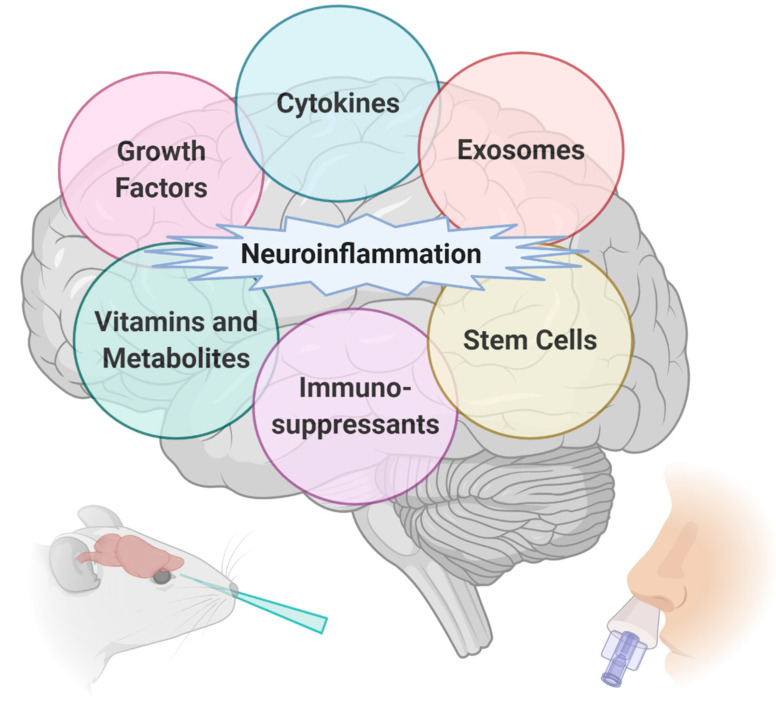
Intranasal (IN) delivery of therapeutics aids in targeting neuroinflammation in many different central nervous system (CNS) diseases such as Alzheimer’s disease, Parkinson’s disease, multiple sclerosis, stroke, and traumatic brain injuries. Generated with BioRender.com.

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
