# Peer review of "Intranasal Delivery: Effects on the Neuroimmune Axes and Treatment of Neuroinflammation"

_pharmaceutics, 2020, doi:10.3390/pharmaceutics12111120_

Round 1

Reviewer 1 Report

In this review, Authors  have highlighted the of many different classes of compounds, focusing on neuroinflammation in CNS diseases including stroke, Traumatic Brain Injuries, Alzheimer Disease, Parkinson’s Disease, Huntington’s disease and Multiple Sclerosis.

Based on large literature Authors showed that Intranasal delivery of many drugs could improve the bioavailability and in a number of cases also the potential therapeutic effect with respect to the oral or intravenous administration and this way is a viable route for the delivery to the brain on neuroimmune modulating substances.

At time, we have only pre-clinical data and in majority in rodents and only sometimes in primates, but the biological evidences are very interesting, in reducing the number of expected side effects, and showing that direct activity on sustained neuroinflammation can play a pivotal role in the above mentioned disease

Minor  concern

In my opinion, the introduction of one or more tables that summarize and organize the main results obtained and the pharmacological differences between intranasal administration and other routes could facilitate the reading and use of the paper.

Page 13 line 570= what means ALS?

Author Response

We thank the reviewer for their careful consideration of the manuscript.

While we agree a table comparing the differences between intranasal delivery and other routes of delivery would be helpful to put the summarized data into context, this would require an extensive review of the literature on the different routes of delivery for every substrate covered (peripheral administration, direct brain administration, etc…). Instead, we have now included two references (line 67 and line 155) on IN delivery that have compared administrative routes. We have also included a table that summarizes most of the substrates we have touched on and their beneficial effect.

Line 575- We removed this acronym (ALS) as we realized we did not touch on it throughout the manuscript and therefore, is out of place in the conclusions.

Reviewer 2 Report

In this paper the authors provide a very thorough and up-to-date review of applications concerning the intranasal delivery of therapeutics. Several examples are reported, ranging from growth factors to cytokines, immunomodulatory compounds to vitamins, exosomes and stem cells.

I found particulalry interesting the comparison of the modalities of administration among different species.

I only have a few suggestions that I believe could help to improve the manuscript:

  1. In the abstract (line 12) the word "substrates" is too generic. Authors should clarify better.
  2. pag.2 line 65. Authors should provide some refernces for this sentence.
  3. I think that the manuscript would benefit if the authors added a table where they list the several compounds tested by IN administration (described in chapters 3 through 8). For each of them, authors could provide a synthetic description of the effects reported upon IN delivery, with main focus on neuroinflammatory paramenters modified by the treatment.

Author Response

We appreciate the reviewer’s constructive comments.

  1. We have rephrased this sentence (line 12) to give a better representation of the work being covered to state “…to use intranasal delivery of various compounds from growth factors to stem cells to reduce neuroimmune interactions”.
  2. We realize we left out an important advantage to IN delivery and that is the relative non-invasiveness of this administrative route. Therefore, our claim on line 65 is further supported as ICV delivery often does not meet this 4th criteria of being non-invasive (line 64). We have also included a reference that has summarized IN delivery compared to other routes of delivery to help support this claim (line 67).
  3. We have included a Table that helps to summarize the effects of each intranasally delivered compound covered in this review (Table 1).